# Immunotherapy in Glioblastoma: Current Shortcomings and Future Perspectives

**DOI:** 10.3390/cancers12030751

**Published:** 2020-03-22

**Authors:** Bas Weenink, Pim J. French, Peter A.E. Sillevis Smitt, Reno Debets, Marjolein Geurts

**Affiliations:** 1Department of Neurology, Erasmus MC Cancer Institute, Be430A, PO Box 2040, 3000 CA Rotterdam, The Netherlands; 2Laboratory of Tumor Immunology, Department of Medical Oncology, Erasmus MC Cancer Institute, 3000 CA Rotterdam, The Netherlands

**Keywords:** adoptive T cell therapy, antigens, checkpoint inhibitors, clinical studies, glioblastoma, immune privilege, tumor micro-environment, vaccines

## Abstract

Glioblastomas are aggressive, fast-growing primary brain tumors. After standard-of-care treatment with radiation in combination with temozolomide, the overall prognosis of newly diagnosed patients remains poor, with a 2-year survival rate of less than 20%. The remarkable survival benefit gained with immunotherapy in several extracranial tumor types spurred a variety of experimental intervention studies in glioblastoma patients. These ranged from immune checkpoint inhibition to vaccinations and adoptive T cell therapies. Unfortunately, almost all clinical outcomes were universally disappointing. In this perspective, we provide an overview of immune interventions performed to date in glioblastoma patients and re-evaluate their performance. We argue that shortcomings of current immune therapies in glioblastoma are related to three major determinants of resistance, namely: low immunogenicity; immune privilege of the central nervous system; and immunosuppressive micro-environment. In this perspective, we propose strategies that are guided by exact shortcomings to sensitize glioblastoma prior to treatment with therapies that enhance numbers and/or activation state of CD8 T cells.

## 1. Introduction

Glioblastomas are the most common type of primary brain cancer affecting 17/500,000 individuals per year. The disease is clinically devastating with patients having a median survival of less than 15 months (hereafter referred to as mo) [1] despite standard-of-care (SOC), which currently consists of surgical resection followed by radiotherapy with concomitant and adjuvant temozolomide (TMZ) chemotherapy. Cancer immunotherapy, whereby one employs and/or activates the host’s own lymphocytes to recognize and destruct cancer, has shown impressive results in the treatment of various extracranial (metastatic) tumor types, e.g., melanoma, non-small cell lung cancer, and non-Hodgkin lymphoma [2,3]. Three major immunotherapeutic approaches comprise immune checkpoint inhibition, vaccination, and adoptive transfer of effector lymphocytes, which will be separately introduced in the sections below. Outcomes of these immunotherapies to treat glioblastoma patients have been rather disappointing and warrant a careful re-evaluation of clinical data in order to justify whether and how to proceed with immune therapies to treat glioblastoma. In this perspective, we will summarize the results from trials of the above-mentioned three types of immunotherapies conducted to date in glioblastoma, followed by a discussion on how current failure of monotherapies in glioblastoma relates to three main determinants of resistance. Improved understanding of key glioblastoma:immune cell interactions will guide the development and testing of novel approaches to sensitize glioblastoma to immune therapies, and we will propose some of these combination treatments that may prove more effective in the future.

## 2. Single Immune Therapies are not Effective in Glioblastoma

### 2.1. Immune Checkpoint Inhibitors

Immune checkpoint inhibitors (ICIs) block inhibitory receptors and their ligands often expressed by immune cells, such as intratumoral T cells and myeloid cells, and tumor cells with monoclonal antibodies to elicit an effective antitumor immune CD8 T cell response. Several so-called ‘immune checkpoints’ have been discovered over the last few decades, including the well-recognized molecules Programmed cell death protein 1 (PD-1), Programmed death-ligand 1 (PD-L1) and Cytotoxic T-lymphocyte-associated protein 4 (CTLA-4). Therapies with these ICIs have shown impressive increases in survival and response rates when compared to chemotherapy in a number of different cancer types [2,4,5,6]. ICIs that have been subsequently approved by the Food and Drug Administration (FDA) and European Medicines Agency (EMA) for the treatment of melanoma, non-small cell lung cancer (NSCLC), renal cell carcinoma (RCC), Hodgkin lymphoma, head and neck squamous cell carcinoma (HNSCC), and urothelial carcinoma include nivolumab and pembrolizumab targeting PD-1, and atezolizumab (for urothelial cancer and NSCLC), avelumab (merkel cell carcinoma and RCC), and durvalumab (NSCLC) targeting PD-L1, as well as ipilimumab targeting CTLA-4 (for melanoma and RCC) [2]. Table 1 lists the results from all phase II and III trials that examined the clinical effect of ICIs in glioblastoma patients.

#### 2.1.1. PD-1 Inhibitor Trials

CheckMate 143 (NCT02017717) was the first phase III trial to evaluate the efficacy of nivolumab +/− ipilimumab versus Vascular Endothelial Growth Factor (VEGF)-inhibitor bevacizumab in patients with recurrent glioblastoma [7]. The primary endpoint of this trial, overall survival (OS), was not improved after treatment with nivolumab (*n* = 184) compared to bevacizumab (*n* = 185), with a median survival time of 9.8 and 10.0 mo, respectively. Although responses were more durable in the nivolumab treatment arm, no data was reported on long-term survivors amongst the entire patient cohort. Another phase III trial, CheckMate 498, which assessed the clinical effect of nivolumab plus radiotherapy versus SOC in patients with newly diagnosed, O-6-methylguanine DNA methyltransferase (MGMT)-unmethylated glioblastoma (*n* = ~275 per arm) also failed to show impact on OS (NCT02617589) [8]. The counterpart of this trial, CheckMate 548 (NCT02667587), in which similar treatments were tested in MGMT-methylated glioblastoma patients (a phase II trial with *n* = ~160 per arm), showed no increase in progression-free survival (PFS). It should be noted, however, that results from CheckMate 548 are still preliminary, and OS data still has to mature before conclusions can be drawn.

A single-arm, phase II trial (NCT02550249) demonstrated the safety of presurgical and postsurgical nivolumab in a set of primary (*n* = 3) and recurrent (*n* = 27) glioblastoma patients. Although no obvious clinical benefit was observed after salvage surgery, two of the primary cases treated with nivolumab showed long-term survival [9]. Patients treated with presurgical and adjuvant pembrolizumab resulted in significant improvement of OS when compared to adjuvant administration alone (median 13.7 mo vs. 7.5 mo) in a different randomized phase II trial for recurrent glioblastoma patients (*n* = 16 vs. 19) [10]. Neoadjuvant immunotherapy may increase the likelihood of targeting tumor-specific T cells as the tumor bulk may be leveraged as a richer source for such T cells [10].

On an anecdotal level, two cases of glioblastomas with a high mutational burden (hypermutated gliomas) resulting from a germline mismatch repair gene mutation have demonstrated dramatic responses to anti-PD-1 therapy [11,12]. However, no tumor responses were observed in four other patients with hypermutated recurrent glioblastomas treated with ICIs [13]. TMZ treatment can also result in tumors with a high mutational load due to acquired mutations in the DNA repair genes MSH2, MSH6, PMS2, and MLH2 [14,15,16,17], and these tumors are likely more immunogenic compared to other glioblastomas. Hypermutation at glioblastoma diagnosis or at recurrence is associated with enhanced numbers of CD8 T cells [18].

#### 2.1.2. PD-L1 Inhibitor Trials

Besides PD-1, preliminary results are available from the first two single-arm phase II studies which tested PD-L1 inhibitors in glioblastoma patients using efficacy as primary endpoints. In the first study conducted in recurrent glioblastoma, the combination of anti-PD-L1 antibody avelumab and axitinib (a tyrosine kinase inhibitor selective for VEGF receptors) did not meet the study threshold for activity (six-mo PFS being 18%) [19]. In the second study, the addition of PD-L1 inhibitor durvalumab to SOC resulted in a modest increase of OS for patients with newly diagnosed MGMT-unmethylated glioblastoma (durvalumab 15.1 mo vs. historical controls 12.7 mo) [20].

### 2.2. Vaccination with Peptides or Dendritic Cells

Tumor vaccines induce a cellular and/or humoral immune response directed against one or multiple tumor antigens. Vaccines generally constitute of peptides (single or combinations) or protein(s) but may also consist of dendritic cells (DCs) either loaded with whole cell tumor lysates or gene-engineered to express a certain (combination of) antigen(s). Immunostimulatory adjuvants (e.g., poly ICLC) are usually co-administered with tumor vaccines to promote adaptive anti-tumor immunity. Completed phase II and III trials using peptide and DC vaccines to target glioblastoma are listed in Table 2 and Table 3.

#### 2.2.1. Peptide Vaccination Trials

In gliomas, the variant III of the epidermal growth factor receptor (EGFRvIII), which results from an in-frame intragenic deletion of EGFR exons 2–7, is present in approximately 20% to 30% of glioblastomas and a recognized target in many peptide vaccination studies [21,22,23]. A peptide vaccine targeting EGFRvIII (rindopepimut) displayed evidence for immunogenicity and efficacy in early phase trials for EGFRvIII+ glioblastoma patients [24,25,26,27]. However, the randomized ACT IV study failed to confirm this initial result and was terminated after interim analysis (mean OS for rindopepimut 20.1 mo versus control 20.0 mo) [28]. The lack of therapeutic effect in ACT IV may be explained by the non-use of historical controls as a comparator (as has been done for the earlier ACT trials) and the observation that EGFRvIII is not homogeneously expressed in all tumor cells, leading to immune escape and recurrence of EGFRvIII-negative tumor cells Additionally, EGFRvIII expression is lost in 30–50% of glioblastoma samples upon tumor recurrence [16,24,29,30]. While rindopepimut and other peptide vaccines for glioblastoma have elicited immune responses, their use has currently not resulted in substantial clinical benefit.

#### 2.2.2. DC Vaccination Trials

So far, one large phase III study has investigated the therapeutic potential of DC vaccination in glioblastoma [31]. After tumor resection and chemoradiotherapy, 331 primary patients were randomized to receive either TMZ and an autologous tumor lysate-pulsed DC vaccine (DCVax-L) or TMZ plus placebo. As all patients were allowed to receive DCVax-L after tumor recurrence, there was a high crossover fraction (~90%) in this trial. The study’s primary endpoint, PFS, has not yet been evaluated and will be the subject of later analyses. The secondary endpoint, OS, has currently been analyzed for the total intention-to-treat patient population and showed a 8-mo survival benefit for the addition of DCVax-L to SOC. Mutation status of isocitrate dehydrogenase (*IDH*) 1 and 2 genes was not determined for this trial, which may affect clinical outcome (as *IDH* mutant glioblastoma patients have a better prognosis). Further maturation of the trial data is required to better assess the efficacy of DCVax-L in newly diagnosed glioblastoma.

### 2.3. Adoptive Transfer of Effector Lymphocytes

Another intervention to provide a tumor-specific immune response is adoptive T cell therapy (AT). In this setup, autologous T cells are expanded in vitro and returned to the patient in large numbers. Early approaches made use of therapeutic T cells that are non-engineered, such as tumor-infiltrating lymphocytes (TILs), while more recently recognized types of AT utilizes engineered (i.e., genetically modified) T cells or other lymphocytes that express either chimeric antigen receptors (CARs) or tumor-specific T cell receptors (TCRs). Table 4 and Table 5 list the performed phase I and II trials with adoptively transferred lymphocytes in glioblastoma.

#### 2.3.1. Gene-Engineered T cells

Only a limited number of small CAR T cell trials for recurrent glioblastoma patients have been carried out so far, and these mainly focused on feasibility and safety of adoptively transferred T cells (Table 4). Importantly, CAR T cells have been shown to migrate from patient peripheral blood to regions of active glioblastoma. In a phase I trial by O’Rourke and colleagues, ten patients with EGFRvIII+ recurrent glioblastoma received a single dose of autologous CAR T cells directed against EGFRvIII [32]. CAR T cell trafficking to the tumor could be verified by histological analysis of surgical specimens obtained after therapy. No objective responses were observed in any of the patients in this trial, and neither in a pilot study testing anti-EGFRvIII CAR T cell treatment in recurrent glioblastoma patients after lymphodepleting chemotherapy [33]. These first negative clinical results with anti-EGFRvIII CAR T cells may be explained by the considerable intratumor, intertumor and temporal heterogeneity of EGFRvIII’s expression in glioblastoma. Indeed, antigen loss was observed in a subset of patients after treatment. Of note, intracranial administration of CAR T cells targeting interleukin-13 receptor alpha 2 (IL-13Rα2) resulted in robust antitumor immunity and complete tumor regression in one patient with recurrent multifocal glioblastoma, which sustained for 7.5 mo following CAR T cell infusions [34]. Currently, no results from TCR T cell therapy trials have been reported for glioblastoma patients. Two phase II studies (NCT03412877, NCT04102436) have recently been initiated by the National Cancer Institute (NCI) that will assess the clinical response after adoptive transfer of T cells genetically engineered to express TCRs reactive against neoantigens in patients with glioblastoma and other (metastatic) cancer types.

#### 2.3.2. Non-Engineered T Cells and Other Lymphocytes

Other adoptive transfer approaches that have been tested in glioblastoma include those with non-engineered lymphocytes, such as TILs, T cell clones derived from peripheral T cells, natural killer (NK) cells and lymphokine-activated killer (LAK) cells (Table 5). The use of TILs, expanded from fresh surgical specimens and locally re-administered together with IL-2, has been attempted in a single small study for recurrent malignant glioma [35]. Although glioblastoma-infiltrating lymphocytes can be successfully expanded in vitro [36], these protocols are generally not yet efficient (e.g., when compared to melanoma) and TILs demonstrated poor function and severe exhaustion [37,38]. Multiple non-overlapping immunosuppressive mechanisms may drastically impair the yield of glioblastoma-infiltrating lymphocytes (see below). During the last decades, early phase trials have not yet demonstrated clear clinical value for both NK- and LAK-based adoptive transfer. A challenge of NK and LAK therapy is the requirement of co-transfusion with immune stimulants, such as IL-2, which may result in significant toxicities [39]. Other early phase trials have focused on autologous cytomegalovirus (CMV)-specific T cells to target glioblastoma [40]. In these studies, CMV-specific T cells were generated in vitro following stimulation of peripheral blood mononuclear cells (PBMC) with peptides derived from CMV phosphoprotein (pp)65. So far, no objective tumor responses have been observed, neither as a single treatment nor when combined with a pp65 mRNA-loaded DC vaccine [41]. It is noteworthy that large comprehensive studies did not provide evidence for the presence of CMV in glioblastomas, questioning its use as target [40,42]

In summary, results from a considerable number of clinical trials testing ICIs, vaccines, and adoptive transfer have been reported for primary and recurrent glioblastoma patients. Unfortunately, no phase III trials have currently shown significant improvement in OS. Although we should await the results from other clinical trials, the so far limited results using PD-1 or PD-L1 inhibitors suggest that alternative approaches should be tested. However, remarkable therapeutic responses have been observed in some studies and the observation of long-term response of a minority of patients to immune interventions demonstrates the potential of these approaches to target glioblastoma. The early and promising results of neo-adjuvant PD-1 blockade warrants larger randomized clinical trials of this regimen in glioblastoma patients. We also recommend testing ICIs in the setting of recurrent TMZ-treated diffuse low-grade gliomas (LGGs), as hypermutation is correlated to the number of TMZ cycles and occurs most often in tumors progressing from lower grade gliomas (up to 50%) [16,17]

The following section will give an up-to-date overview of important mechanisms that relate to glioblastoma immunotherapy resistance.

## 3. Immune Therapy Resistance in Glioblastoma

Here, we describe the pathophysiology underlying the immune therapy resistance of glioblastoma in relation to three key issues: low immunogenicity; immune privilege of the central nervous system (CNS); and immune suppressive micro-environment.

### 3.1. Low-Immunogenicity of Glioblastoma

The mutational burden and consequently neoantigen burden in glioblastoma is generally low compared to other immunogenic tumor types, such as melanoma and non-small cell lung cancer [43,44]. Different studies have predicted the presence of only a handful of low expressed glioma neoantigens derived from mutated genes [44,45,46,47]. While other, non-mutated targets (e.g., Cancer Germline Antigens (CGAs)) are frequently expressed in glioblastoma, expression levels are low or, at best, highly variable [44,48]. Besides gene expression, tumor antigens need to pass through the antigen processing and presenting machinery (APM) to enable recognition by T cells. Strikingly, a large-scale immunopeptidome study did not identify any human leukocyte antigen (HLA)-presented mutated peptides nor peptides derived from frequently targeted CGAs in glioblastoma patient serum (*n* = 142) or tumor tissue (*n* = 10) [49]. A different study showed that HLA class I and II molecules are not detectable in ~50% or more of glioblastoma patient tissues (*n* = 47) and expression of tapasin, an APM protein that mediates interaction between MHC class I molecules and the transporter associated with antigen processing (TAP), was downregulated in a subset of patient specimens [50]. Other components of the APM, such as TAP1/2 and β2-microglobulin, were still intact in these specimens. We advocate in-depth studies into the APM and immunogenicity of glioblastoma since the presence of a single immunogenic target antigen may potentially already suffice to start an antitumor immune response [51]

### 3.2. Immune Privilege of CNS

Under homeostatic conditions, leukocytes are not present in the brain parenchyma. There are, however, small numbers of immune cells, including T cells, in the choroid plexus stroma and the cerebrospinal fluid, as well as the subarachnoid and the perivascular spaces [52]. Microglia are the most prominent immune cells in the brain and they serve as tissue-resident macrophages of the CNS. In fact, microglia, together with macrophages and dendritic cells in the meninges and perivascular spaces, form the first line of defense against pathogens [53]. Despite their absence in the brain under homeostatic conditions, T cells are able to extravasate from blood vessels in the case of a primary malignant brain tumor. The number of intratumoral CD8 T cells in glioblastoma remains, however, small (0–12% of all cells) [54] when compared to extracranial tumor types [55]. In line with other cancers, CD8 T cell numbers have been reported to be associated with favorable outcome and survival in glioblastoma [56,57,58]. CD8+ T cell numbers, in combination with neoantigen quality, can also predict a subgroup with longest survival [59]. The abundance and tumor invasiveness of CD8 T cells in glioblastoma is higher than in LGGs, which is accompanied by an increased expression of chemo-attractants CXCL9, CXCL10 and intercellular adhesion molecule (ICAM)1 [44]. Nevertheless, a substantial part of CD8 T cells display a PD-1^+^, LAG-3^+^, TIGIT^+^, CD39^+^, KLRG1^−^, and CD57^−^ profile, indicative of T cell exhaustion [37,60] and suggestive that factors in the glioblastoma micro-environment negatively control an anti-tumor CD8 T-cell response.

### 3.3. Immune-Suppressive Micro-Environment

The glioblastoma micro-environment is hostile towards an effective anti-tumor immune response, thereby negatively impacting the effect of immune therapies [61]. Here, we explicitly zoom in on the major cell types and mediators reported to contribute to an immune-suppressive micro-environment in glioblastoma. This sections aims to describe and discuss the role that these defined cells and mediators in the glioblastoma micro-environment potentially play in suppressing immunotherapy for glioblastoma. More comprehensive reviews that elaborate on other cell types and mediators present in the glioblastoma micro-environment are published elsewhere [62,63].

#### 3.3.1. Immune-Suppressive Cells

Tumor associated macrophages (TAMs) are highly prevalent in glioblastoma, and up to 50% of the glioblastoma micro-environment consists of these cells [64]. The number and function of TAMs are shaped by glioblastoma-derived soluble factors, such as the chemo-attractant CCL-2 and colony-stimulation factor 1 (CSF-1) [65]. These cells have a pleiotropic capability to suppress CD8 T cell activity in glioblastoma due to, at least in part, surface-expression of IL-4Rα and the production of arginase and inducible nitric oxide synthase (iNOS). Activation of IL-4Rα up-regulates the expression of transforming growth factor (TGF)-β (see below for immune-suppressive actions of TGF-β), whereas arginase and iNOS deplete key amino acids from the extracellular environment, which in turn inhibits T cell proliferation [66].

Regulatory T cells (Tregs) are a subset of CD4 T cells that generally express the transcription factor Foxp3. The proportion of CD4 cells being Tregs is highly variable in glioblastoma (i.e., 4–55%) [67,68]. Tregs are immune-suppressive as they secrete TGF-β and IL-10, which by limiting T cell interleukin-2 and interferon (IFN)-γ production, result in hampered function of CD8 T cells [37]. Glioblastomas attract Tregs from the periphery via soluble factors, such as tumor cell-derived CCL-22 [69]. Once within the tumor tissue, Tregs can become activated via TAMs that express T-cell immunoglobulin and mucin domain–containing molecule 4 (TIM4). TIM4-expressing myeloid cells can phagocytose tumor-specific T cells that express phosphatidylserine (PS; a possible measure of dying cells), which results in the expression of the immune-suppressive aldehyde dehydrogenase and TGF-β, which in turn stimulate the activity of Tregs [37]. In addition, indoleamine 2,3-dioxygenase (IDO; discussed further below) produced by dendritic cells in tumor-draining lymph nodes, recruits and activates Tregs in glioblastoma [69].

#### 3.3.2. Immune-Suppressive Mediators

As already mentioned above, TGF-β is a pleiotropic cytokine that plays a central role in immune suppression in the glioblastoma micro-environment [70]. TGF-β has three isoforms, and all are (over-) expressed in glioblastoma [71]. TGF-β suppresses IL-2 dependent T cell survival, and further impairs cytotoxic T cells’ activity via inhibiting the expression of immune-stimulatory and effector molecules, such as IL-6, IL-10, IFN-γ, granzymes A and B, perforin, and Fas ligand [11]. Additionally, it drives development of naive T cells into Tregs [72].

Glioblastomas take up and metabolize tryptophan. High IDO activity leads to the depletion of tryptophan from the local micro-environment, and activation of the amino acid starvation-sensing response pathway involving the broadly expressed general control non-derepressible kinase (GCN2). This pathway is an important immune modulator as activation thereof in T cells leads to anergy and subsequent cell death [73]. Alternative routes of tryptophan catabolism via tryptophan-2,3-dioxygenase (TDO) may be of special interest in glioblastoma, as this gene is highly expressed in glioblastoma [73].

## 4. Strategies to Sensitize Glioblastoma to Immune Therapies

The disappointing results of clinical trials on immunotherapy in glioblastoma as discussed in previous sections, reflect the shortcomings of the current immunotherapeutic treatment strategies. Strategies to enhance the sensitivity of glioblastomas to immune therapies are clearly needed. These strategies could address the low-immunogenicity of glioblastoma, attempt to increase the CD8 T cell influx and activity, and/or strive to downregulate the immunosuppressive micro-environment (Figure 1).

### 4.1. Strategies to Enhance the Efficacy of ICI

The strategies that are currently investigated to enhance the efficacy of ICI in glioblastoma address two key points: downregulating the immune suppressive micro-environment and increasing the CD8 T cell influx.

#### 4.1.1. Downregulating the Immune Suppressive Micro-Environment

Many attempts have been undertaken to target TAMs in glioblastoma. These attempts have particularly focused on CSF-1R inhibition as this receptor strongly mediates recruitment of TAMs into tumor tissue [65]. Indeed, combination therapies of TAM-targeting and ICI were shown to increase the number of CD8 T-cells in the tumor in pre-clinical models in melanoma, leading to an increased rejection of tumors [74]. Pre-clinical data of this approach in glioblastoma are lacking, but two phase I trials using this combination therapy are now ongoing: Cabiralizumab, an anti-CSF-1 receptor monoclonal antibody, in combination with nivolumab, is tested in solid cancers, including glioblastoma patients (NCT02526017). Additionally, BLZ945, a CSF-1 inhibitor, is tested as monotherapy and in combination with spartalizumab, a novel anti-PD1 monoclonal antibody, in solid cancers, including glioblastoma (NCT02829723).

Early attempts to deplete Tregs in glioblastoma made use of the constitutive expression by these cells of the high affinity IL-2 receptor α (IL-2Rα/CD25). In a phase I clinical trial in glioblastoma, treatment with the anti-IL-2Rα monoclonal antibody daclizumab in combination with EGFRvIII peptide vaccination, significantly reduced the frequency of circulating Tregs [75]. This study result warrants a follow-up study to investigate the beneficial effect of daclizumab treatment in glioblastoma. Another potential Treg target is the costimulatory receptor glucocorticoid-induced TNFR related protein (GITR). Indeed, intra-tumoral treatment with anti-GITR antibody leads to depletion of Tregs in a glioma murine model [76]. In addition, treatment with anti-GITR plus anti-PD1 antibodies in a murine ovarian cancer model led to dramatic increases in the numbers of CD4 and CD8 T cells with a concomitant decrease in Tregs and myeloid-derived suppressor cells in these tumors [77]. In line with these preclinical results, a phase II study testing the combination of a GITR antibody (INCAGN1876), an anti-PD1 antibody and stereotactic radiosurgery for recurrent glioblastoma has just opened (NCT04225039).

The small molecule inhibitor galunisertib (LY2157299 monohydrate) targets the serine-threonine kinase domain of TGF-βRI, abrogates the phosphorylation of SMAD2, the initial intracellular signaling protein of the TGF-β pathway. In a phase II study (*n* = 158), no clinical benefit was found when comparing galunisertib to lomustine chemotherapy and to galunisertib plus lomustine chemotherapy in recurrent glioblastoma [78]. Moreover, of the 127 tumor specimens examined, no difference between immune cells or immune mediators was found before and after treatment. However, in the patients who were treated with galunisertib only, there was a trend towards an association between the number of galunisertib cycles and the CD4 and lymphocyte cell counts [79]. These data should be interpreted with caution due to the small patient number but do suggest that TGF-β blockade renders tumors susceptible to immune therapies. At the moment of this writing, there are no ongoing studies with TGF-β blockade in glioblastoma.

Monotherapies with IDO inhibitors are currently being explored in glioblastoma (NCT02052648); no clinical trials in glioblastoma are ongoing with TDO-inhibitors. Interestingly, treatment with the IDO-1 enzyme inhibitor BGB-5777, combined with anti-PD1 antibody, as well as radiotherapy in a mouse model of glioblastoma, increased the absolute numbers of CD8 T cells in the tumor and increased survival, when compared to monotherapy with either single agent [80]. The combination of an IDO-1 inhibitor (epacodostat) and ICI is currently under investigation in a phase I trial assessing solid cancers, including glioblastoma (NCT02327078).

#### 4.1.2. Increasing the CD8 T Cell Influx

There is an increasing amount of evidence that oncolytic viruses can be used to sensitize glioblastoma to immune therapies. In fact, it has even been speculated that the immune stimulatory effect of oncolytic viruses is more relevant than their direct oncolytic effects. For example, in an experimental glioma mouse model using oncolytic Newcastle disease virus (NDV), intratumoral administration was associated with increased immunogenicity, elevated numbers of CD8 T cells and reduced accumulation of TAMs in 50% of the treated mice [81]. A notable benefit of combining oncolytic virotherapy and ICI came from preclinical studies in which one tested: measles virus and anti-PD-1 antibody; vesicular stomatitis virus (VSV) and anti-PD-1 antibody; and adenovirus Delta-24-RGDOX (expressing the immune co-stimulatory OX40 ligand) and anti-PD-L1 antibody, as well as reovirus and anti-PD-1 antibody, which all demonstrated therapeutic benefits against the GL261 glioma model [82]. In phase I trials where patients with recurrent glioblastoma were treated with intratumoral adenovirus DNX-2401 [83] or PVSRIPO (a modified polio virus) [84], imaging showed an increase in tumor size and atypical contrast enhancement after treatment in a subset of patients, which has been interpreted as an inflammatory-mediated response. Indeed, histologic analysis of a resected lesion identified inflammatory-mediated responses with high numbers of macrophages and CD8 T cells [83]. The three (12%) responders in the DNX-2401 trial had 10- to 1000-fold increases in interleukin-12p70 in their sera, which is recognized for its induction towards T helper 1 responses and cell-mediated immunity. Currently, it is unclear whether the intra-tumoral CD8 T cells expressed immune checkpoints, which would advocate the additional use of ICI next to vaccinations. A phase 2 clinical trial with DNX-2401 plus pembrolizumab is currently recruiting patients with recurrent glioblastoma (NCT02798406).

As mentioned in previous sections, neoadjuvant ICI may sensitize glioblastoma to other immune therapies. Indeed, molecular analyses revealed enhanced expression of chemo-attractants (i.e., CXCL10, CCL4, and CCL3L1) and presence of T cells in the tumor when compared with a historical group of glioblastoma samples. Interestingly, T cell receptor clonality analyses also showed more diversity in the group treated with nivolumab with an association between TCR clonotype diversity and survival [9]. A gene signature related to IFN-γ responsiveness and a decrease in the number of tumors with cell cycle gene expression signatures was noted in a separate neoadjuvant ICI study [10]. Both studies suggest neoadjuvant treatment could be further harnessed therapeutically via combination with other therapies.

Radiotherapy is another potential glioblastoma sensitizer to ICI, as it opens the blood-brain-barrier, increases the amount of TILs and up-regulates expression of PD-L1 in mice with intracranial gliomas [85]. Additionally, tumor cell death induced by radiotherapy releases tumor-associated antigens into the micro-environment, leading to increased antigen presentation [86]. Treatment with fractionated (high dose) radiotherapy has been shown to up-regulate the expression of PD-L1 in in vitro glioblastoma cell lines [87], and anti-PD1 treatment is effective particularly in combination with radiotherapy in glioma models [85]. In patients, the efficacy and side effects of hypofractioned radiation therapy still have to be determined. Combination therapies with (hypofractionated) radiation and ICI in primary glioblastoma (NCT02313272, NCT02648633, NCT02829931, NCT02866747) are open and currently recruiting patients.

Treatment with a COX-2 inhibitor, which blocks IL-6, IL-10 and GM-CSF in glioma mouse models, ultimately leading to increased numbers of intra-tumoral T cells, may provide yet an additional approach to sensitize glioblastoma to ICI [88].

Low-dose chemotherapy also increases the influx of CD8 T cells into the tumor in pre-clinical models of glioblastoma [89], but the peripheral lymphodepletion in patients treated with chemotherapy may counteract these beneficial effects [79]. Additionally, TMZ administration has been shown to have a detrimental effect on the formation of a memory CD8+ T cell response against glioblastoma [90]. This makes (low dose) chemotherapy unlikely to be an attractive sensitizer for anti-glioblastoma T cell treatment.

### 4.2. Strategies to Enhance the Efficacy of Vaccinations

Since tumor-restricted antigens are scarce, and many are not shared by a large number of patients, personalized vaccine approaches may be of particular interest for the treatment of glioblastoma. Recently, safety and feasibility results from two single arm, open label, phase I trials were reported that both targeted patient-specific repertoires of neoantigens in newly diagnosed glioblastoma. In the first study, the vaccine NeoVax targeted multiple mutated peptides that were selected based on predicted HLA-binding and mRNA expression of the source antigen in tumor tissue (*n* = 8) [47]. The other study (*n* = 15) conducted by the Glioma Actively Personalized Vaccine Consortium (GAPVAC) utilized a similar approach but also included mass spectrometry analysis of patient tumor tissues to select whether mutated peptides are indeed presented by HLA molecules and included unmutated peptides to complete their vaccines [46]. The preliminary median OS data of the GAPVAC-101 trial are encouraging (29.0 mo) and argue that larger and better controlled clinical trials need to be executed.

### 4.3. Strategies to Enhance the Efficacy of Adoptive Transfer of Effector Lymphocytes

Much of the potential of adoptive T cell therapy for glioblastoma lies in modification of the cells themselves [91]. The new generation CAR T cells are armed with immune stimulatory cytokines that improve CAR T cell expansion and persistence, while rendering them resistant to the immune suppressive tumor micro-environment. In glioblastoma studies, CAR T cells targeting IL-13Rα2 were modified to over-express transgenic IL-15 and demonstrated that IL-15 cytokine secretion was T cell activation dependent and resulted in improved CAR T cell persistence in vitro. This also translated into significantly improved anti-tumor activity in vivo [92].

Gene editing to further enhance the efficacy of CAR-T cells is still in its very early days, especially in glioblastoma. The knock-out of the intracellular signaling molecule diacylglycerol kinase (DGK) using a CRISPR/Cas9-based strategy in EGFRvIII CAR T cells led to a significantly less sensitivity to TGF-β mediated suppression and did not show significant loss of effector function following repeated stimulation. These in vitro findings translated to in vivo studies in a mouse glioma model as tumor-bearing mice receiving double knock-out EGFRvIII CAR T cells had significantly reduced tumor burden with increased frequencies of tumor-infiltrating lymphocytes [93]. The-knock in of genes to enhance CAR T cell function has not yet been tested in glioblastoma [91].

Multitarget engineered T-cells for glioblastoma to overcome inter- and intratumor cell antigen heterogeneity are quickly entering the field. Trivalent CAR T cells targeting three glioblastoma associated antigens (HER2, IL-13Rα2-, and EphA2-specific CAR molecules, all expressed simultaneously on the T cell surface) are now designed [94]. Even more recently, other antigens have been targeted including CSPG4 [95] or B7-H3 [96] and chlorotoxin [97] as has been eloquently demonstrated in preclinical studies. These proof-of-concept studies are promising; however, the question remains if these CAR T cell products will be efficient when tested clinically [91]. Due to the variation in antigen expression in glioblastoma, target selection on a patient-by-patient basis may be necessary.

### 4.4. Combination of ICI, Vaccination, and Adoptive Transfer of Effector Lymphocytes

In a trial using neoantigen-targeting vaccines, two out of 8 vaccinated patients had vaccine-specific T-cell responses [47]. In these patients, thorough immune monitoring revealed that vaccine-induced T cells (predominantly CD4 T cells) were able to migrate to the tumor site in the brain. These T-cells predominantly had an exhausted phenotype and expressed multiple immune checkpoints, suggesting the option of these vaccines combining with ICI [47]. In a second peptide vaccination trial, using both non-personalized and personalized vaccines, 12 out of 13 patients treated with the non-personalized vaccine had CD8 T cells that recognized at least one of the vaccine protein, and 8 out of 10 patients treated with the personalized vaccine had CD4 T-cells that recognized neoantigens. Clinical trials combining a peptide vaccine plus a checkpoint inhibitor in glioblastoma are currently ongoing (NCT02529072, NCT03014804).

The effector functions of CAR-T cells have previously been shown to be enhanced by ICI in glioma models [98]. Currently, phase I studies combining anti-EGFRvIII CAR-T cells with pembrolizumab (NCT03726515) and anti-IL-13Rα2 CAR-T cells with or without ipilimumab and nivolumab (NCT04003649) are now recruiting patients with glioblastoma.

## 5. Outstanding Questions for Future Research

Rational combinatorial approaches will be required in order to achieve optimal efficacy of immunotherapy in patients with glioblastoma. Many unanswered questions remain. First, most patients with glioblastoma are treated with dexamethasone, and the immune suppressive effects of this drug inevitably intervene with immunotherapy trials. For example, in a trial using neoantigen-targeting vaccines [47], T-cell responses were induced only in the patients who did not receive dexamethasone in the vaccine priming phase.

Second, the treatment of CNS pathology in general is limited by poor drug penetration of the blood-brain-barrier, which is significantly prohibitive for compounds of sizes greater than 400–600 Da [99]. The calculated molecular mass of nivolumab, which is administered in the peripheral blood, is 146 kDa. Although the blood-brain-barrier is disrupted in glioblastoma, it is unknown to what extent systemically administered antibodies such as nivolumab reach the tumor site. Of interest, patients with brain metastases of melanoma do respond to systemically administered nivolumab [100], which suggests that the blood-brain-barrier is not per se limiting towards treatment effect. We argue that pharmacokinetic assessments should be included in future studies. In fact, modern imaging technologies may facilitate the assessment of drug delivery, and radioisotopes coupled to, among others, monoclonal antibodies may aid drug delivery [101].

Third, accurate predictive biomarkers to select patients that will benefit from the immunotherapeutic treatment are obviously needed. Even in the absence of benefit for the total patient group, there may very well be a subset of patients that does respond. For example, 8% of patients responded to nivolumab in the CheckMate 143 trial of recurrent glioblastoma [7]. Along these lines, it is highly encouraging that the design of clinical trials for glioblastoma is more and more taking into account the collection of tumors before and after treatment [9,10,79].

## 6. Conclusions

Results of conducted randomized trials on immunotherapy in glioblastoma inevitably, and almost collectively, point to a lack of efficacy. These disappointing results may reflect the shortcomings of the current immunotherapeutic treatment strategies. Glioblastomas are very low-immunogenic tumors, located in a T-cell poor CNS compartment, and embedded in a particularly immune suppressive micro-environment. We, therefore, recommend future treatment strategies that sensitize glioblastoma to immune therapies. Rationally designed combinatorial immunotherapeutic approaches offer tremendous opportunities to ultimately fulfill the high promises of immunotherapy in glioblastoma.

## Figures and Tables

**Figure 1 cancers-12-00751-f001:**
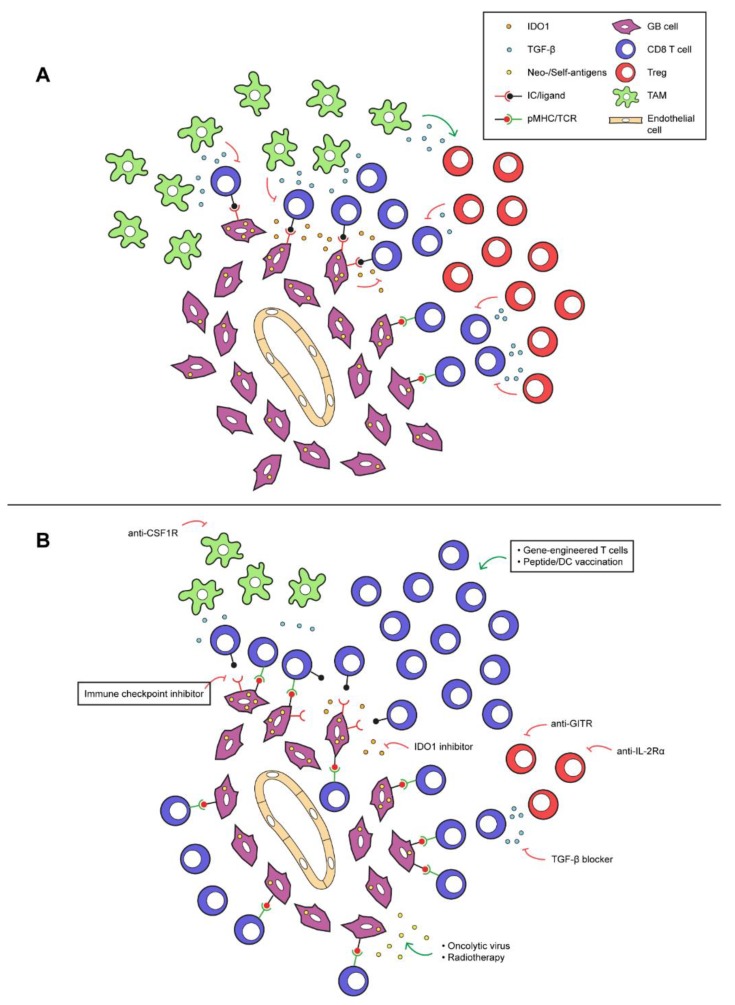
Exemplary immune-suppressive mechanisms in glioblastoma, and therapeutic interventions to sensitize glioblastoma for T cell treatments. (**A**) Tumor-associated macrophages (TAMs) and regulatory T cells (Tregs) release immunosuppressive mediators in the glioblastoma microenvironment, such as TGF-β and IL-10 (the latter not depicted). Local production of indoleamine 2,3- dioxygenase (IDO) or tryptophan-2,3-dioxygenase (TDO) by glioblastoma cells depletes tryptophan from the tumor micro-environment, having an adverse effect on the function of CD8 T cells. Also, the immune checkpoint, PD-L1, expressed on glioblastoma cells can engage with PD-1+ T cells to suppress their effector function. (**B**) Monoclonal antibodies directed against glucocorticoid-induced TNFR related protein (GITR) and IL-2 receptor α (IL-2Rα) may specifically deplete intratumoral Tregs. Immunosuppressive effects exerted by TGF-β can be abolished by small molecule inhibitors, such as galunisertib. IDO and TDO inhibitors (the latter not depicted) may restore tryptophan levels in the micro-environment, causing re-activation of CD8 T cells. TAM-induced immunosuppression may be counteracted using inhibitors against colony-stimulation factor 1 (CSF1) or its receptor. Oncolytic virotherapy and radiotherapy may cause increased release of antigens, which in turn may result in enhanced numbers and activation of intratumoral CD8 T cells. When (one or several of) these interventions are combined with vaccines, immune checkpoint inhibitors (ICIs) or adoptive transfer of effector lymphocytes, this may significantly improve the recruitment and/or activation state of glioblastoma-specific CD8 T cells and lead to restoration of the efficacy of these T cell treatments in glioblastoma.

**Table 1 cancers-12-00751-t001:** Phase II/III clinical trials with immune checkpoint inhibitors in glioblastoma.

Clinical Trial	Phase	Target	Treatment	Control	Indication	# Patients	Endpoint	Outcome
CheckMate 143 NCT02017717 PMC5463583	III	PD-1	Nivolumab	Bevacizumab	R GB	Each arm: ~185	OS	No impact
CheckMate 498 NCT02617589	III	PD-1	RT, nivolumab	SOC	P GB MGMT-unmeth.	Each arm: ~275	OS	No impact
CheckMate 548 NCT02667587	II	PD-1	SOC, nivolumab	SOC, placebo	P GB MGMT-meth.	Each arm: ~160	PFS	No impact
Neo-nivo NCT02550249 30742120	II	PD-1	Nivolumab (neo-adjuvant), (S)S, nivolumab (adjuvant)	None	P GB R GB	P: 3 R: 27	(OS)	7.3 mo
NCT02337491	II	PD-1	Pembrolizumab, bevacizumab	Pembrolizumab	R GB	Treatment: 50 Control: 30	(OS)	8.8 mo vs. 10.3 mo
NCT02337686	II	PD-1	SS, pembrolizumab	None	R GB	15	PFS6	53%
NCT02852655 30742122	II	PD-1	Pembrolizumab (neo-adjuvant), SS, pembrolizumab (adjuvant)	SS, pembrolizumab (adjuvant)	R HGG	Treatment: 16 Control: 19	(OS)	13.7 mo vs. 7.5 mo
NCT03291314	II	PD-L1	Avelumab, axitinib	None	R GB	32	PFS6	18%
SEJ NCT03047473	II	PD-L1	SOC, avelumab	None	P GB	24	(PFS)	11.9 mo
NCT02336165	II	PD-L1	SOC, durvalumab	Historical	P GB MGMT-unmeth.	40	OS12	60% vs. 50%

Abbreviations: MO, month; #, number of; RT, radiotherapy; SOC, standard of care; SS, second surgical procedure; P, primary; R, recurrent; GB, glioblastoma; MGMT, O6-methylguanine-DNA methyltransferase gene; (un)meth., (un)methylated gene promoter; PFS6, 6-mo progression-free survival (PFS); OS12, 12-mo OS. Reference numbers (NCT, PMID, PMCID) corresponding to each clinical trial are indicated in the ‘Clinical trial’ column. The ‘Endpoint’ column indicates which primary survival endpoint was assessed in each clinical trial. If a study only used a secondary survival endpoint, the outcome measure was placed between parentheses.

**Table 2 cancers-12-00751-t002:** Phase II/III clinical trials with peptide vaccines in glioblastoma.

Clinical Trial	Phase	Target	Treatment	Control	Indication	# Patients	Endpoint	Outcome
HSPPC-96 NCT00905060	II	Autologous peptides	SOC, PEP	None	P GB	46	OS	23.8 mo
HSPPC-96 NCT00293423 24335700	II	Autologous peptides	SS, PEP	None	R GB	41	OS6	90.2%
HSPPC-96 NCT01814813	II	Autologous peptides	PEP Bevacizumab	Bevacizumab	R GB	30	OS	No impact
ACT-IV NCT01480479 28844499	III	EGFRvIII	SOC, PEP	SOC, KLH	P GB EGFRvIII+	371	OS	No impact
ACT-III 25586468	II	EGFRvIII	SOC, PEP	Historical	P GB EGFRvIII+	65	PFS5.5	66% vs. 45%
ACT-II 21149254	II	EGFRvIII	SOC, PEP	Historical	P GB EGFRvIII+	22	(OS)	23.6 vs. 15.0 mo
ReACT NCT01498328	II	EGFRvIII	PEP Bevacizumab	KLH Bevacizumab	R GB EFGRvIII+	33	PFS6	27% vs. 11%
ACTIVATe NCT00643097 20921459	II	EGFRvIII	SOC, PEP	Historical	P GB EGFRvIII+	18	PFS6	94% vs. 59%
ITK-1 UMIN000006970 30500939	III	Multiple TAA	PEP	Placebo	R GB HLA-A24+	Treatment: 58 Control: 30	OS	No impact
SL-701 NCT02078648	II	Multiple TAA	PEP Bevacizumab	None	R GB HLA-A2+	74	OS12	43%
IMA-950 NCT01920191 30753611	I/II	Multiple TAA	SOC, PEP	None	P GB HLA-A2+	16	(OS)	19 mo
SurVaxM~ NCT02455557	II	Survivin	SOC, PEP	None	P GB HLA-A2, -A3, -A11 and -A24+	55	OS12	70.8%

Abbreviations: TAA, tumor-associated antigen; PEP, peptide vaccination; KLH, Keyhole limpet hemocyanin; SS, second surgical procedure; P, primary; R, recurrent; GB, glioblastoma; HLA, human leukocyte antigen; #, number of; PFS5.5, 5.5-mo PFS; OS6, 6-mo OS; PFS6, 6-mo PFS; OS12, 12-mo OS. The ‘Endpoint’ column indicates which primary survival endpoint was assessed in each clinical trial. If a study only used a secondary survival endpoint, the outcome measure was placed between parentheses.

**Table 3 cancers-12-00751-t003:** Phase II/III clinical trials with dendritic cells (DC) vaccines in glioblastoma.

Clinical Trial	Phase	Loading Material for DCs	Treatment	Control	Indication	# Patients	Endpoint	Outcome
NCT01567202 30159779	II	GSC antigens	P: SOC, DC R: SS, RT/CT, DC	P: SOC, placebo R: SS, RT/CT, placebo	P GB R GB	Treatment: 22 Control: 21	PFS	7.7 mo vs. 6.9 mo
NCT02772094 21715171	I/II	Irradiated tumor cells	SOC/SS, DC	None	P GB	16	OS	17 mo
ICT-107 NCT01280552 31320597	II	Multiple TAA	SOC, DC	SOC, placebo	P GB HLA-A1+ and/or -A2+	Treatment: 81 Control: 43	OS	17.0 vs. 15.0 mo
DCVax-L NCT00045968 29843811	III	Tumor lysate	SOC, DC	SOC, placebo	P GB	331	(OS)	23.1 mo
GBM-Vax NCT01213407 30301187	II	Tumor lysate	SOC, DC	SOC	P GB	34	PFS12	No impact
NCT03879512 30054667	I/II	Tumor lysate	SS, Cyclophosphamide, DC	None	R GB pediatric R GB adult	Pediatric: 6 Adult: 5	OS6	100%
DEND/GM NCT01006044 28499389	II	Tumor lysate	SOC, DC	None	P GB	31	PFS	12.7 mo
NCT00576537 18632651	I/II	Tumor lysate	SOC/SS, DC	None	P GB R GB	P: 11 R: 23	(OS)	Vaccine responders: 21 mo Nonresponders 14 mo
NCT00323115 21499132	II	Tumor lysate	SOC, DC	None	P GB	10	(OS)	28 mo
DENDR2 NCT02820584	I/II	Tumor lysate	(1) SS, TMZ, DC (2) SS, TTX, DC	None	R GB	(1) 12 (2) 8	(OS)	(1) 7.4 mo (2) 9.2 mo
DENDR1 29632727	I/II	Tumor lysate	SOC, TMZ, DC	None	P GB	24	PFS12	41%
DC-CAST-GBM NCT00846456 23817721	I/II	Tumor stem cell mRNA	SOC, DC	None	P GB	7	(OS)	25 mo

Abbreviations: GSC, glioma stem cell; TAA, tumor-associated antigen; P, primary; R, recurrent; GB, glioblastoma; #, number of; SS, second surgical procedure; RT, radiotherapy; CT, chemotherapy; TTX, tetanus toxoid; OS6, 6-mo OS; PFS12, 12-mo PFS. The ‘Endpoint’ column indicates which primary survival endpoint was assessed in each clinical trial. If a study only used a secondary survival endpoint, the outcome measure was placed between parentheses.

**Table 4 cancers-12-00751-t004:** Phase I clinical trials with gene-engineered T cells in glioblastoma.

Clinical Trial	Phase	CAR Generation	Target	Other Treatment	Indication	# Patients	OR (%)	CR (%)
NCT00730613 26059190	I	First CD3z	IL13Rα2	SS	R GB	3	0/3 **(0)**	0/3 **(0)**
NCT02208362 28029927	I	Second 41BB-CD3z	IL13Rα2	SS	R GB IL13Rα2+	1	1/1 **(100)**	1/1 **(100)**
NCT01109095 28426845	I	Second CD28-CD3z	HER2 CMV pp65	SS	R GB HER2+	17	1/17 **(6)**	0/17 **(0)**
NCT02209376 28724573	I	Second 41BB-CD3z	EGFRvIII	SS	R GB EGFRvIII+	10	0/10 **(0)**	0/10 **(0)**
NCT01454596 30882547	I	Third CD28-41BB-CD3z	EGFRvIII	Cyclophosphamide, Fludarabine, IL-2	R GB EGFRvIII+	18	0/18 **(0)**	0/18 **(0)**

Abbreviations: SS, second surgical procedure; IL-2, interleukin 2; R, recurrent; GB, glioblastoma; OR, objective response; CR, complete response; both according to Response Assessment in Neuro-Oncology (RANO) criteria. Number of patients with responses = before dash. Total number of patients treated = after dash. Percentage of responses = between brackets (bold).

**Table 5 cancers-12-00751-t005:** Phase I/II clinical trials with non-gene engineered T cells and other lymphocytes in glioblastoma.

Clinical Trial	Phase	Lymphocytes	Other Treatment	Indication	# Patients	OR (%)	CR (%)	OS
9390198	I	Alloreactive CTLs	SS, IL-2	R A, O grade III-IV	5	0/5 **(0)**	0/5 **(0)**	-
24795429	I	Autologous CMV-specific T cells	CT	R GB CMV seropositive	11	0/11 **(0)**	0/11 **(0)**	-
NCT01588769 29511178	I	Autologous CTLs and NK cells	None	R GB	10	3/10 **(30)**	0/10 **(0)**	-
9647171	I	Autologous T lymphocytes	Irradiated tumor cell vaccination	R A grade III-IV	10	3/10 **(30)**	0/10 **(0)**	-
NCT00004024 16817692	II	Autologous T lymphocytes	Irradiated tumor cell vaccination	R A or O grade II-IV	19	8/19 **(42)**	1/19 **(5)**	-
10778730	I	Autologous TILs	SS, IL-2	R GB	6	3/6 **(50)**	1/6 **(17)**	-
NCT00331526 19816190	II	Autologous LAK cells	SOC, IL-2	P GB	33	NR	NR	20.5 mo
NCT00003067 8625188	I	Autologous LAK cells	SS, IL-2	R GB R A grade III	19	4/19 **(21)**	2/19 **(11)**	-

Abbreviations: SS, second surgical procedure; IL-2, interleukin 2; CT, chemotherapy; CTL, cytotoxic T lymphocyte; P, primary; R, recurrent; GB, glioblastoma; A, astrocytoma; O, oligodendroglioma; NR, not reported; OR, objective response; CR, complete response; both according to Response Assessment in Neuro-Oncology (RANO) criteria. Number of patients with responses = before dash. Total number of patients treated = after dash. Percentage of responses = between brackets (bold).

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
