# Peer review of "Immunotherapy in Glioblastoma: Current Shortcomings and Future Perspectives"

_cancers, 2020, doi:10.3390/cancers12030751_

Round 1
Reviewer 1 Report
This is a very helpful summary reviewing the clinical, therapeutic and immunological situation.
Some suggestions.
Tables.
Please consider the possibility to add a column containing the references for each study.
It should be better to include SOC (standard of care) in the table legend (at least Table 1.)
Note that the study DENDR1 in table 3 includes not only SOC but also TMZ as an adjuvant (in the column “treatment” the authors have to indicate SOC, TMZ, DC)
TILs
(200-205) Please consider the observation that appeared in contrast with (Ref 35, and Mohme (Mohme et al., 2018). Specifically, Liu and colleagues reported that TILs isolated from GBM could be expanded in vitro (Liu et al., 2017). In another study from Prins group, next-generation deep sequencing of the TCRVβ CDR3 region was performed to determine presence, frequency, and phenotype of TIL in GBM after DC vaccination showing that TIL content in GBM can predict a better survival (Hsu et al., 2016) and is correlated with better patient survival.
CD8 T
Other studies have described the importance of CD8 T cell infiltration.
Wang and colleagues support that CD8+ T cells are enriched in TMZ-induced hypermutated (recurrent) GBM (Wang et al., 2018). CD8+ T cells infiltration, in combination with neoantigen quality, can also predict a subgroup with longest survival
CAR-T
CAR T cells for adoptive transfer have been developed also for glioblastoma (GBM) with specificity for IL13Rα2, HER2, EGFRvIII and EphA2 and have been evaluated in preclinical studies with encouraging results. More recently other antigens have been targeted including CSPG4 or B7-H3.
Very recently it has been shown that CAR T cells can be efficiently directed against chlorotoxin as a targeting domain (Wang et al., 2020).
Minor point
399-402. In DENDR1 study TMZ, used as an adjuvant, was found to play a negative role in CD8+ T cells in treated patients failing to generate a memory status (Pellegatta et al., 2018).
Reviewer 2 Report
The manuscript is structured and to some extent analytical but not comprehensive.
1) In the the sectiom "3.3.1. Immune-suppressive cells", the authors mention largely tumor associated macrophages (TAMs) and regulatory T cells (Tregs). However, neutrophils, MDSCs, immunosuppressive Th cells (CD3+CD4+FoxP3− type 1 regulatory T cells ) are ignored.
see also Z. Li, X. Liu, R. Guo, P. Wang, CD4+Foxp3- type 1 regulatory T cells in glioblastoma multiforme suppress T cell responses through multiple pathways and are regulated by tumor-associated macrophages., Int. J. Biochem. Cell Biol. 81 (2016) 1–9. doi:10.1016/j.biocel.2016.09.013.
2) In the section "3.3.2. Immune-suppressive mediators", the authors mention only TGF-β, tryptophan and high IDO activity. However, a number of immune-suppressive mediators is overwhelming, which should be listed/mentioned, at least partially.
3) Figure 1 has no legend
Reviewer 3 Report
This manuscript is a timely review covering some important topics and issues regarding immunotherapy in glioblastomas. The manuscript in general, is well-written, insightful and fairly comprehensive. I only have a few suggestions for the authors’ consideration to further improve the quality of the manuscript.
- The authors mentioned that immune-suppressive micro-environment contributes to the immune therapy resistance in glioblastoma and discussed several important immune-suppressive mediators including TGF-beta. They also should talk about the role of COX/PGE2-mediated inflammatory processes in immunosuppression and treatment resistance in glioblastomas, as there are abundant evidences (Please see summary in PMID: 27693715).
- It is understood that the major focus of this review is glioblastomas, but I think the author may also want to briefly talk about the immunotherapy in low-grade glioma for comparison with glioblastomas.
- Why are the immune therapies (monoclonal antibodies) not effective in glioblastoma? Does their poor brain-penetration play a role? Page 13, line 464, not sure why the authors state “which should enter the brains following systemic administration”. The molecular mass of nivolumab is 146 Kilo Da, which is way more than 400-600 Da that the authors described as a criteria for brain-penetration of therapeutic compounds.
